# A Combination of Sorafenib, an Immune Checkpoint Inhibitor, TACE and Stereotactic Body Radiation Therapy versus Sorafenib and TACE in Advanced Hepatocellular Carcinoma Accompanied by Portal Vein Tumor Thrombus

**DOI:** 10.3390/cancers14153619

**Published:** 2022-07-25

**Authors:** Zeyu Zhang, Chan Li, Weijun Liao, Yun Huang, Zhiming Wang

**Affiliations:** 1Department of Hepatobiliary Surgery, Xiangya Hospital, Central South University, Changsha 410078, China; zeyu_zhang1994@csu.edu.cn (Z.Z.); 198112258@csu.edu.cn (W.L.); 402128@csu.edu.cn (Z.W.); 2Department of Cardiovascular Medicine, Xiangya Hospital, Central South University, Changsha 410078, China; chanli@csu.edu.cn

**Keywords:** downstaging, hepatocellular carcinoma, sorafenib, stereotactic body radiation therapy, transcatheter arterial chemoembolization, immune checkpoint inhibitor

## Abstract

**Simple Summary:**

Portal vein tumor thrombosis (PVTT) is the commonest type of macrovascular invasion in hepatocellular carcinoma (HCC), while the effectiveness of treatments for HCC with PVTT remains unsatisfactory. The present study aimed to assess the effectiveness of the combination treatment of sorafenib, an immune checkpoint inhibitor, transcatheter arterial chemoembolization and stereotactic body radiation therapy in patients with advanced HCC and PVTT. We confirmed that the combination therapy yielded better survival data than the combined administration of sorafenib and transcatheter arterial chemoembolization in patients with advanced HCC and PVTT. Especially, this combination therapy could serve as a downstaging strategy to provide the chance of radical surgery for the patients with advanced HCC.

**Abstract:**

Background: This study compared the effectiveness of the combined administration of sorafenib, an immune checkpoint inhibitor, transcatheter arterial chemoembolization (TACE), and stereotactic body radiation therapy (SBRT) (SITS group) vs. sorafenib combined with TACE (ST group) in treating and downstaging advanced hepatocellular carcinoma (HCC) with portal vein tumor thrombus (PVTT). Methods: The present study included patients with advanced HCC and PVTT treated with one of the above combination therapies. The downstaging rate, objective response rate (ORR), progression-free survival (PFS), overall survival (OS), disease control rate (DCR), and adverse events (AEs) were assessed. Results: Sixty-two patients were analyzed. The ORR was elevated in the SITS group compared with the ST group (*p* = 0.036), but no differences were found in DCR (*p* = 0.067). The survival analysis revealed higher PFS (*p* = 0.015) and OS (*p* = 0.013) in the SITS group, with median PFS and OS times of 10.4 and 13.8 months, respectively. Ten patients displayed successful downstaging and underwent surgery in the SITS group, vs. none in the ST group. The prognosis was better in surgically treated patients compared with the non-surgery subgroup, based on PFS (*p* < 0.001) and OS (*p* = 0.003). Despite a markedly higher rate of AEs in the SITS group (*p* = 0.020), including two severe AEs, the SITS combination therapy had an acceptable safety profile. Conclusion: The SITS combination therapy yields higher PFS and OS than the combined administration of sorafenib and TACE in patients with advanced HCC and PVTT, especially as a downstaging strategy before surgery.

## 1. Introduction

Remarkable advances have been made in the treatment of hepatocellular carcinoma (HCC) in the past few years, including radiotherapy, transarterial chemoembolization (TACE), hepatic arterial infusion chemotherapy, targeted therapy, and immunotherapy [1]. Nevertheless, the overall prognosis remains unsatisfactory, especially in patients with HCC and macrovascular invasion, which is widely considered an unfavorable prognostic factor [2].

Portal vein tumor thrombosis (PVTT) is the commonest type of macrovascular invasion in HCC, followed by hepatic vein tumor thrombus, inferior vena cava tumor thrombus, and atrial tumor thrombus [3]. The literature has shown a reduced survival of patients with HCC and PVTT compared with those without PVTT [4]. Based on the Barcelona Clinic for Liver Cancer (BCLC) staging system, HCC with PVTT is categorized into BCLC stage C, indicating advanced HCC [5]. At present, no worldwide protocol guiding the treatment of HCC with PVTT is available. Therapies such as TACE, chemotherapy, radiotherapy, use of tyrosine kinase inhibitors (TKIs), immune checkpoint inhibitors (ICIs), and their combinations can be potentially used for treating advanced HCC, and many researchers are making great efforts to identify the most effective combination [6].

We previously reported that combined treatment with sorafenib, camrelizumab, TACE, and stereotactic body radiation therapy (SBRT) downstaged advanced HCC with PVTT [7]. A mean progression-free survival (PFS) of 15.7 months was reported, with an overall response rate reaching 41.7%. Most importantly, four of twelve patients were downstaged and could undergo radical surgery. Nevertheless, the effect of this combination on patient outcomes after surgery was not fully investigated by the previous case series study.

Thus, the present study aimed to assess the effectiveness of the above combination treatment in patients with advanced HCC and PVTT. Moreover, the role of surgery in this combination therapy was preliminarily investigated.

## 2. Methods

### 2.1. Patients

This retrospective study examined patients with advanced HCC and PVTT treated with the combination of sorafenib, an immune checkpoint inhibitor (camrelizumab or tislelizumab), TACE, and SBRT (SITS group), or with the combination of sorafenib and TACE (ST group) in Xiangya Hospital, Central South University, between January 2014 and August 2021. HCC diagnosis was made using the noninvasive criteria from the European Association for the Study of Liver (EASL) guidelines [2]. HCC staging was based on the Barcelona Clinic Liver Cancer (BCLC) system, while PVTT assessment used Cheng’s PVTT classification [8]. The exclusion criteria were: HCC recurrence, previous antitumor therapies such as surgery and systematic therapy, distant metastasis, Child–Pugh grade C, and hepatitis C virus or human immunodeficiency virus infection. The study was approved by the ethics committee of the Xiangya Hospital of Central South University. The requirement for informed consent was waived by the committee since the data were assessed retrospectively.

### 2.2. Study Design

In the SITS group, oral sorafenib (Bayer and Onyx, Shanghai, China) was administered at 400 mg BID, which was decreased to 200 mg BID in case of intolerable drug-related adverse events (AEs). Intravenous tislelizumab (BeiGene Co., Ltd., Beijing, China) or camrelizumab (Jiangsu Hengrui Medicine Co., Lianyungang, China) was given at 200 mg at 3-week intervals. TACE was carried out by superselective cannula placement in the artery supplying the lesion and with lipiodol (Jiangsu Hengrui Medicine Co., Lianyungang, China) and cisplatin (Qilu Pharmaceutical Co., Ltd., Jinan, China) injection. SBRT was performed with a CyberKnife (Accuray, Sunnyvale, CA, USA) with 36 to 42 Gy in 4- to 5-Gy fractions. Sorafenib and ICIs were started simultaneously, and TACE and SBRT were carried out within 2 weeks and 1 month, respectively. Maintenance with sorafenib and ICI was performed until serious AE occurrence, tumor progression, or surgical procedure following successful downstaging.

### 2.3. Data Collection and Definitions

Continuous follow-up was carried out monthly upon treatment initiation. Blood tests, e.g., hepatic function and tumor biomarker assessments, and computed tomography (CT) were performed at 3-month intervals. MRI was performed as needed. Tumor response, downstaging rate, PFS, overall survival (OS), objective response rate (ORR), disease control (DCR) rate, and AEs were examined. PFS was determined from treatment initiation to disease progression or patient death. OS was determined from treatment initiation to patient death. ORR was the proportion of patients who achieved complete response (CR) or partial response (PR). DCR represented the proportion of individuals achieving CR or PR, or stable disease (SD) as the best response. Follow-up ended on 30 October 2021. The modified response evaluation criteria in solid tumors (mRECIST) were used for tumor response evaluation. The National Cancer Institute’s Common Terminology Criteria for Adverse Events (NCI-CTCAE) v5.0 were used to define AEs. Successful downstaging was reflected by absent tumoral arterial enhancement of PVTT based on the mRECIST and EASL criteria [9,10]. The secondary surgery was conducted once the successful downstaging was observed, and the sorafenib and ICI administrations were still performed after the secondary surgery.

### 2.4. Statistical Analysis

The data were assessed with SPSS v23.0 (SPSS, New York, NY, USA) and GraphPad (GraphPad Prism Software, San Diego, CA, USA). Continuous and categorical variables were shown as means ± standard deviation (SD) and number (%), respectively. Kaplan–Meier curve analysis was carried out to determine PFS and OS, which were compared by the Gehan–Breslow–Wilcoxon test.

## 3. Results

### 3.1. Patient Characteristics

The study flowchart is depicted in Figure 1.

Sixty-two patients were analyzed (Table 1), including 56 men (90.3%). They were 50.4 ± 12.8 years old. Most patients (58/62) had previous hepatitis B infection, and 51 (82.3%) had liver cirrhosis. Tumor sizes were 9.3 ± 3.8 cm, and 33 patients (53.2%) showed multiple lesions. Based on Cheng’s PVTT classification, there were 7, 30, and 25 Type I, II, and III cases, respectively. Liver function in most patients (56/62, 90.3%) remained intact. Tislelizumab was used in 8 patients and camrelizumab in 22 cases. Overall, no differences were detected in patient and tumor characteristics between the two groups.

### 3.2. Effectiveness of the Combination Therapies

Treatment effects are summarized in Table 2. PR, SD, and disease progression were observed in ten (33.3%), seven (23.3%), and seven (23.3%) patients of the SITS group, respectively, vs. eight (25.0%), nine (28.1%), and fifteen (46.9%) in the ST group, respectively. Complete response was achieved in six patients in the SITS group vs. none in the ST group. The ORR was significantly higher in the SITS group than in the ST group (53.3% vs. 25.0%, *p* = 0.036), but no differences were found in DCR (*p* = 0.067). Importantly, twelve (40.0%) patients experienced successful downstaging after SITS administration, vs. none among ST-treated cases. Survival analysis also showed longer median PFS (10.4 vs. 6.3 months, *p* = 0.015) and OS (13.8 vs. 8.8 months, *p* = 0.013) in SITS-treated cases compared with patients administered ST (Figure 2).

### 3.3. Univariable and Multivariable COX Regression Analyses

In order to examine the role of the combination therapy in HCC patients, univariable and multivariable COX regression analyses were performed for PFS (Table 3) and OS (Table 4). The results confirmed that SITS combination therapy independently predicted PFS and OS in patients with advanced HCC. In addition, the presence of multiple tumors independently predicted PFS. Child–Pugh classification independently predicted both PFS and OS.

### 3.4. Role of Secondary Surgery in the SITS Combination Therapy

In this study, twelve patients experienced successful downstaging, which was defined as absent tumoral arterial enhancement of PVTT based on the mRECIST and EASL criteria, but two refused surgery. In order to examine the role of secondary surgery in the SITS combination therapy, preliminarily, we subdivided the SITS group into the surgery (*n* = 10) and non-surgery (*n* = 20) subgroups. As shown in Figure 3, the prognosis was significantly better in surgically treated individuals compared with the non-surgery subgroup in terms of PFS (20.6 vs. 7.4 months, *p* < 0.001) and OS (24.2 vs. 12.1 months, *p* = 0.003), but the different prognosis might not result from surgery but from the effectiveness of the SITS combination therapy itself. However, given the principle of beneficence, surgery must be suggested to every patient who experienced a successful downstaging; as a result, the number of patients (2) who experienced successful downstaging without secondary surgery was too small to analyze.

### 3.5. AEs of the Combination Therapy

Two patients in the SITS group developed intolerable skin reactions and discontinued therapy. There were no subsequent readmissions or deaths in either group. A significantly higher rate of AEs was detected in the SITS group (76.7% vs. 46.9%, *p* = 0.020). The most common AE was fever (18/62, 29.0%), followed by skin reactions (14/62, 22.6%) and fatigue (6/62, 9.7%), ranked. Overall, the safety profile of the SITS combination therapy was acceptable (Table 5).

## 4. Discussion

Our previous case series study developed a novel combination treatment comprising sorafenib, ICI, TACE, and SBRT for advanced HCC accompanied by PVTT. This comparative study further investigated the effectiveness of the SITS combination therapy. Compared with the combined administration of sorafenib and TACE, the SITS combination therapy showed significantly higher effectiveness and better patient prognosis, with an acceptable safety profile. More importantly, 40% of the patients administered with the SITS combination therapy experienced successful downstaging, and most of them underwent curative surgery. This downstaging strategy might offer a chance of curative surgery for improved prognosis in advanced HCC cases accompanied by PVTT.

For approximately ten years, advanced HCC was systemically managed only with the TKI sorafenib. With the development of multiple immunotherapies, combination therapies involving TKIs and ICIs currently have potent therapeutic effectiveness in advanced HCC [11], but the effects of these combination therapies remain far from satisfactory. Efforts have also been made to combine systemic therapies with locoregional treatments such as TACE, microwave ablation (MVA), and SBRT [12]. Wu et al. combined TACE and sorafenib to treat advanced HCC, showing better effectiveness than sorafenib alone [13], which was used for comparison in the present study. Chen et al. showed that the combined treatment with apatinib, TACE, and MWA is effective in BCLC stage C HCC [14]. Studies also attempted to examine ICIs for their roles in combination therapies. Combined treatment consisting of TACE, ablation, apatinib, and camrelizumab was reported in advanced HCC with promising results and acceptable safety profiles [15]. This work reported an SITS combination treatment comprising sorafenib, ICI, TACE, and SBRT in advanced HCC with PVTT. Synergistic effects have been reported between TKI and ICI [16]. Several pathways have been identified that may synergize with immunotherapy, including MAPK [17], VEGF [18], WNT/b-catenin [19], and PTEN/PI3K [20] pathways. Moreover, other work has demonstrated that molecules involved in epigenetic modifications can modulate tumor antigen expression [21] or suppress T-cell infiltration in tumors [22,23] and, therefore, comprise additional candidates for combinations with checkpoint inhibitors. SBRT has also been reported to have synergistic effects by combining with immunotherapy [24]. Radiation therapy may upregulate the expression of both antigen-presenting cells as well as effector T cells and increase overall T-cell infiltration into tumors, and thus enhancing the effects of immunotherapies [25]. Furthermore, locoregional treatments might enhance local hypoxia and vascular permeability, induce immunogenicity via the release of tumor antigens following cancer cell death, and elicit damage-associated molecular patterns, which might be rescued by TKIs and ICIs [26]. It has also been reported that SBRT is particularly effective in tumor areas with high oxygenation, namely the tumor periphery, where TACE itself is less effective [27]. Meanwhile, cytotoxic agents used for TACE could lead to a higher radiosensitivity [28], supporting the combination of TACE and SBRT. Taken together, we encourage researchers and clinicians to determine the most effective combination with appropriate patient selection.

Ten patients experienced tumor downstaging and underwent curative surgery in this study. Although salvage surgery after tumor downstaging is believed to be beneficial in advanced HCC [29], the role of surgery in this combination therapy has not been well investigated. The better patient prognosis might not result from surgery but from the effectiveness of the SITS combination therapy itself, because the patients who can receive a secondary surgery (downstaging achieved) already had a better response to the combination therapy than the patients who cannot receive a secondary surgery (downstaging not achieved). However, surgery must be suggested to every patient with successful downstaging, according to medical ethics; as a result, only a few patients experienced tumor downstaging without subsequent salvage surgery. Therefore, determining the role of surgery in this combination therapy remains challenging. Another potential issue for the SITS therapy is patient selection. Although the present study showed that almost half of the enrolled advanced HCC patients might benefit from the SITS therapy, a proper patient selection may even raise the ORR. With the growing number of enrolled patients, biomarker analysis and second-generation sequencing may be helpful to predict responders and non-responders in future studies. Recently, tumor mutation burden was reported to have a significant correlation with clinical outcomes of ICI [30], which may serve as another potential biomarker in response to combination therapies in the treatment of HCC. In addition, a study by Sia et al. revealed a subgroup of HCCs with markers of the inflammatory response, such as fewer chromosomal aberrations, and markers of cytolytic activity might be susceptible to immunotherapy [31]. Overall, the discovery of effective predictive biomarkers for patient selection may be the next milestone in the field of both immunotherapies and associated combination therapies.

## 5. Conclusions

The SITS combination treatment comprising sorafenib, ICI, TACE, and SBRT has better effectiveness than combined treatment with sorafenib and TACE in advanced HCC accompanied by PVTT, especially as a downstaging strategy. AEs showed a markedly elevated rate, but the safety profile remained acceptable.

## Figures and Tables

**Figure 1 cancers-14-03619-f001:**
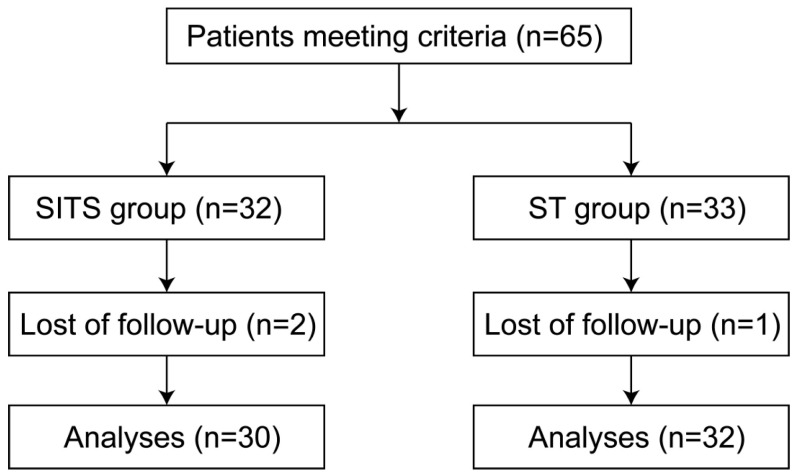
Study workflow.

**Figure 2 cancers-14-03619-f002:**
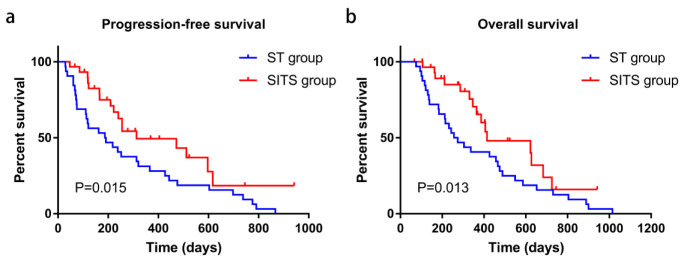
Kaplan–Meier curve analysis of PFS (**a**) and OS (**b**) in both patient groups. PFS, progression-free survival; OS, overall survival.

**Figure 3 cancers-14-03619-f003:**
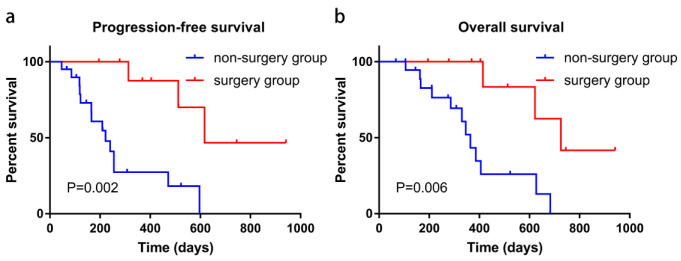
Kaplan–Meier curve analysis of PFS (**a**) and OS (**b**) in the surgery and non-surgery subgroups. PFS, progression-free survival; OS, overall survival.

**Table 1 cancers-14-03619-t001:** Clinical and pathological data.

Characteristic	ST Group (*n* = 32)	SITS Group (*n* = 30)	*p*-Value
Age (years)	51.00 (40.50, 61.00)	52.00 (43.25, 59.25)	0.555
Gender			0.418
Male	30 (93.8)	26 (86.7)	
Female	2 (6.3)	4 (13.3)	
HBsAg			0.613
Positive	29 (90.6)	29 (96.7)	
Negative	3 (9.4)	1 (3.3)	
Liver cirrhosis			0.101
Yes	29 (90.6)	22 (73.3)	
No	3 (9.4)	8 (26.7)	
Tumor size (cm)	7.85 (6.10, 11.63)	9.00 (7.00, 13.28)	0.302
Number of tumors			0.799
Single	14 (43.8)	15 (50.0)	
Multiple	18 (56.2)	15 (50.0)	
Cheng’s PVTT classification			0.367
I	5 (15.6)	2 (6.7)	
II	13 (40.6)	17 (56.7)	
III	14 (43.8)	11 (36.7)	
AFP (ng/mL)			0.311
≤400	15 (46.9)	10 (33.3)	
>400	17 (53.1)	20 (66.7)	
Albumin (g/L)	37.90 (33.80, 42.38)	39.90 (34.90, 42.63)	0.423
Total Bilirubin (μmol/L)	15.55 (10.95, 29.45)	15.50 (10.38, 21.75)	0.073
Prothrombin Time (s)	13.95 (12.38, 14.78)	13.05 (12.28, 13.73)	0.130
Child–Pugh classification			0.197
A	27 (84.4)	29 (96.7)	
B	5 (15.6)	1 (3.3)	
ICI			-
Camrelizumab	-	22 (73.3)	
Tislelizumab	-	8 (26.7)	

Data are median (interquartile range) or *n* (%). HBsAg, hepatitis B surface antigen; PVTT, portal vein tumor thrombus; AFP, alpha-fetoprotein; ICI, immune checkpoint inhibitor.

**Table 2 cancers-14-03619-t002:** Therapeutic efficacy.

Characteristic	ST Group (*n* = 32)	SITS Group (*n* = 30)	*p*-Value
Best overall response			0.020
Complete response	0 (0.0)	6 (20.0)	
Partial response	8 (25.0)	10 (33.3)	
Stable disease	9 (28.1)	7 (23.3)	
Progressive disease	15 (46.9)	7 (23.3)	
Objective response rate	8 (25.0)	16 (53.3)	0.036
Disease control rate	17 (53.1)	23 (76.7)	0.067
Successful downstaging	0 (0.0)	12 (40.0)	-

Data are *n* (%).

**Table 3 cancers-14-03619-t003:** Univariable and multivariable analyses of risk factors for PFS in advanced HCC cases accompanied by PVTT.

Variable	PFS	
	HR (95% CI)	*p*-Value
Univariable analysis		
Age (years)	0.979 (0.956, 1.002)	0.072
Gender (male vs. female)	1.437 (0.560, 3.686)	0.451
HBsAg (positive vs. negative)	1.268 (0.390, 4.120)	0.693
Liver cirrhosis (yes vs. no)	0.800 (0.544, 1.177)	0.257
Tumor size (cm)	1.066 (0.987, 1.150)	0.103
Number of tumors (multiple vs. single)	1.828 (1.028, 3.250)	0.040
Cheng’s PVTT classification		
II vs. I	0.663 (0.282, 1.560)	0.346
III vs. I	0.928 (0.388, 2.221)	0.868
AFP (>400 vs. ≤400)	1.269 (0.711, 2.265)	0.420
Albumin (g/L)	1.025 (0.972, 1.082)	0.360
Total Bilirubin (μmol/L)	1.009 (0.980, 1.037)	0.555
Prothrombin Time (s)	0.873 (0.699, 1.090)	0.229
Child–Pugh classification (B vs. A)	2.396 (1.001, 5.740)	0.049
Combination therapy (SITS vs. ST)	0.522 (0.288, 0.946)	0.032
Multivariable analyses		
Age (years)	0.985 (0.961, 1.010)	0.251
Number of tumors (multiple vs. single)	2.241 (1.147, 4.375)	0.018
Child–Pugh classification (B vs. A)	3.373 (1.308, 8.697)	0.012
Combination therapy (SITS vs. ST)	0.462 (0.252, 0.845)	0.012

PFS, progression-free survival; HCC, hepatocellular carcinoma; PVTT, portal vein tumor thrombus; HR, hazard ratio; CI, confidence interval; HBsAg, hepatitis B surface antigen; AFP, alpha-fetoprotein.

**Table 4 cancers-14-03619-t004:** Univariable and multivariable analyses of risk factors for OS in advanced HCC cases accompanied by PVTT.

Variable	OS	
	HR (95% CI)	*p*-Value
Univariable analyses		
Age (years)	0.984 (0.961, 1.008)	0.192
Gender (male vs. female)	1.154 (0.722, 1.842)	0.550
HBsAg (positive vs. negative)	1.095 (0.336, 3.563)	0.881
Liver cirrhosis (yes vs. no)	0.817 (0.542, 1.231)	0.334
Tumor size (cm)	1.072 (0.989, 1.161)	0.092
Number of tumors (multiple vs. single)	1.708 (0.944, 3.092)	0.077
Cheng’s PVTT classification		
II vs. I	0.673 (0.285, 1.586)	0.365
III vs. I	0.828 (0.340, 2.018)	0.678
AFP (>400 vs. ≤400)	1.341 (0.735, 2.447)	0.338
Albumin (g/L)	1.028 (0.971, 1.090)	0.342
Total Bilirubin (μmol/L)	1.008 (0.979, 1.038)	0.591
Prothrombin Time (s)	0.876 (0.699, 1.099)	0.253
Child–Pugh classification (B vs. A)	2.524 (1.051, 6.062)	0.038
Combination therapy (SITS vs. ST)	0.541 (0.290, 1.008)	0.053
Multivariable analyses		
Tumor size (cm)	1.087 (0.994, 1.190)	0.068
Number of tumors (multiple vs. single)	1.880 (0.935, 3.782)	0.077
Child–Pugh classification (B vs. A)	5.241 (1.878, 14.628)	0.002
Combination therapy (SITS vs. ST)	0.478 (0.256, 0.893)	0.021

OS, overall survival; HCC, hepatocellular carcinoma; PVTT, portal vein tumor thrombus; HR, hazard ratio; CI, confidence interval; HBsAg, hepatitis B surface antigen; AFP, alpha-fetoprotein.

**Table 5 cancers-14-03619-t005:** Adverse events.

Characteristic	ST Group (*n* = 32)	SITS Group (*n* = 30)	*p*-Value
Incidence of adverse events	15 (46.9)	23 (76.7)	0.020
Fever	8	10	
Skin reaction	7	7	
Fatigue	2	4	
Diarrhea	0	3	
Vomiting	1	3	
Reduction of platelet	0	2	
Hypertension	2	2	
Headache	0	2	
Level III or IV adverse events	0	2 (6.7)	-
Readmission for adverse events	0	0	-
Death due to adverse events	0	0	-

Data are *n* (%).

## Data Availability

All data generated or analyzed during this study are included in this published manuscript.

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
