# Peer review of "A Combination of Sorafenib, an Immune Checkpoint Inhibitor, TACE and Stereotactic Body Radiation Therapy versus Sorafenib and TACE in Advanced Hepatocellular Carcinoma Accompanied by Portal Vein Tumor Thrombus"

_cancers, 2022, doi:10.3390/cancers14153619_

Round 1

Reviewer 1 Report

This study was reported the utility of the combination therapy for HCC with portal vein thrombus. The reviewer would like to suggest some critiques as follows.

1.     The author should help of a native English speaker prior to submit the manuscript and make more concise this manuscript.

2.     The authors used the following word: the current work and the present study. which is more appropriate?

3.     “experienced successful downstaging” is unclear. The authors should describe the definition of this point.

4.     “wile great effort … effective position.” Is unclear.

5.     The authors should use the median and IQR instead of mean and SD.

6.     On page 2, line 12, space is deleted between present and study.

Author Response

1. Thank you for your suggestion. This manuscript has been revised by a native English speaker and the language editing certification has also been uploaded for review.

2. Thank you for your suggestion. We have changed “the current work” to “the present study” throughout the manuscript to avoid misunderstanding.

3. We are sorry for our ambiguous wording. We have described the definition ofsuccessful downstaging in the Methods section and the Results section of the revised manuscript.

“Successful downstaging was reflected by absent tumoral arterial enhancement of PVTT based on the mRECIST and EASL criteria [9, 10].

4. We are sorry for our ambiguous wording. We have revised the sentence according to your suggestion.

“many researchers are making great efforts to identify the most effective combination.”

5. Thank you for your suggestion. We have used the median and IQR according to your suggestion.

6. Thank you for your noticing. We have revised our manuscript according to your suggestion.

Reviewer 2 Report

The authors compare the effectiveness of the SITS combination therapy, which comprises of sorafenib, an immune checkpoint inhibitor (tislelizumab or camrelizumab), transcatheter arterial chemoembolization (TACE) and stereotactic body radiation therapy (SBRT), with combined administration of sorafenib and TACE (ST group) for treating advanced hepatocellular carcinoma (HCC) accompanied by portal vein tumor thrombus (PVTT). Their results showed that the combination of SITS improved overall response rate (ORR, 53.3% versus 25.0%, P=0.036), overall survival (OS, (13.8 versus 8.8 months, P=0.013) and median progression-free survival (PFS, 10.4 versus 6.3 months, P=0.015) than ST group in advanced HCC cases accompanied by PVTT. Furthermore, 40% of patients administered SITS combination therapy experienced a successful downstaging. Overall, the manuscript is well written, and the experimental data are well-presented, but some details need to be addressed before publication.

I have the following comments for this manuscript:

1. In the study design part, please specify the supplier of the sorafenib, camrelizumab, lipiodol and cisplatin. In addition, please clarify when the secondary surgery was conducted and whether sorafenib and ICI administrations were performed after surgery.

2. In Figure 2, please adjust the font size of the figure title to make it consistent with others, as the “

progression-free survival; OS, overall survival” is bigger than others.

3. On Page 6, in the “3.4. The role of secondary surgery in the SITS combination therapy” part, please correct the “prognoses” in the sentence “Different prognoses might not result from surgery, but from the effectiveness of the SITS combination therapy itself”. In addition, please indicate the PFS and OS for surgery and non-surgery group in the main text. Furthermore, 12 of patients administered SITS combination therapy experienced a successful downstaging, 10 of which received curative surgery and resulted in increased PFS and OS. Please explain how the authors come to conclude that “Different prognoses might not result from surgery, but from the effectiveness of the SITS combination therapy itself”.

Author Response

1. Thank you for your suggestion. We have specified the supplier of the sorafenib(Bayer and Onyx, Shanghai, China), camrelizumab(Jiangsu Hengrui Medicine Co., Jiangsu, China), lipiodol (Jiangsu Hengrui Medicine Co., Jiangsu, China) and cisplatin (Qilu Pharmaceutical Co., Ltd, Shandong, China).

The secondary surgery was conducted once the successful downstaging was observed, and the sorafenib and ICI administrations were still performed after the secondary surgery, which has been described in the revised manuscript as you suggested.

2. Thank you for your noticing. We have revised our manuscript and figures according to your suggestion.

3. Thank you for your noticing. We have revised our manuscript according to your suggestion. The PFS and OS for the surgery and the non-surgery grouphave been indicated in the Results part of the revised manuscript.

When we tried to explain the better prognosis results of patients who received surgery, we realized that the better prognosis results might not come from the surgery, because the patients who can receive a secondary surgery (downstaging achieved) already had a better response to the combination therapy than the patients who can not receive a secondary surgery (downstaging not achieved). However, we must suggest surgery to every patient with successful downstaging according to medical ethics, leading to the low number of patients who experienced tumor downstaging without subsequent salvage surgery. Thus, the role of secondary surgery still needs to be further investigated in future studies.

We are sorry that we poorly explained it in the original manuscript, and we have further explained and discussed it in the Results and Discussion parts of the revised manuscript.

Round 2

Reviewer 1 Report

The authors revised the paper in accordance with the reviewers’ comments.  The reviewer believes that this paper will provide useful information for readers.

Reviewer 2 Report

The authors have satisfactorily responded to all of my comments and made necessary changes to the manuscript. As technical points have been clarified, no major points left.